# Textual Prototypes Guided Balanced Visual Feature Learning For Long-Tailed Vision Recognition

## Abstract

In recent advancements, pre-trained contrastive models like CLIP have demonstrated remarkable multi-modal prowess in tackling diverse vision tasks. Yet, their potential in addressing the long-tailed vision recognition challenge has not been thoroughly investigated. In this study, we observe that textual features coming from CLIP exhibit a more discriminative and balanced distribution compared to their visual counterparts. Leveraging this insight, we propose a novel approach that uses these balanced textual features as prototypes to guide the learning of robust, disentangled representations from biased visual features. Our method begins with the fine-tuning of CLIP through contrastive learning, enabling the encoders to better adapt to the target dataset. Subsequently, we freeze the visual encoder and apply a linear adapter to enhance the visual representations. To achieve robust vision recognition, we integrate a linear classifier into our framework, which is initialized with the fine-tuned textual features and the weights can be viewed as prototypes. We then introduce a principled approach to robust vision representation learning by minimizing the optimal transport distance between the refined visual features and the prototypes, facilitating the disentanglement of biased features and the iterative optimization of prototypes towards class centroids. Additionally, we introduce a supervised contrastive learning loss based on the transport plan for further enhanced robust vision representation learning. Extensive experiments on long-tailed vision recognition benchmarks demonstrate the superiority of our method.

## 1 Introduction

The triumph of deep learning is largely dependent on high-quality large-scale datasets (Russakovsky et al., 2015), which usually assume a nearly balanced data distribution. However, real-world datasets typically exhibit a long-tailed distribution, where only a few head classes encompass most of the data and most tail classes own very few samples (Zhou et al., 2017; Lin et al., 2014; Liu et al., 2015; Asuncion, 2007). This phenomenon poses a significant challenge for deep learning models, resulting in poor performance on tail classes (Zhou et al., 2020a; Liu et al., 2019; Yang et al., 2022). Consequently, addressing this issue is of paramount practical importance.

Typically, methods designed to solve the long-tailed recognition can be roughly divided into five approaches: data-level approach (Shen et al., 2016; Barandela et al., 2004; Han et al., 2005; Chawla et al., 2002), re-weighting approach (Liu et al., 2022; Lin et al., 2017; Cui et al., 2019; Ren et al., 2018; Cao et al., 2019; Guo et al., 2022a), meta-learning approach (Guo et al., 2022a; Li et al., 2021; Ren et al., 2020; Jamal et al., 2020), decoupling approach (Kang et al., 2019; Zhong et al., 2021) and others (Menon et al., 2020; Wang et al., 2021a; Li et al., 2022b; Wang et al., 2021b; Long et al., 2022; Dong et al., 2022). Despite their effectiveness, these methods only utilize the image modality to solve the long-tailed classification problem. That is to say, they ignore the semantic features of the raw label text. Considering the language modality might provide beneficial information specially for the inadequate data sources, we can address the long-tailed problem based on vision-language models, where the representation power of the vision-language model is the key. Recently, contrastive vision-language model such as CLIP (Radford et al., 2021) has provided an effective way to learn powerful visual-language representations. Motivated by this, some methods (Ma et al., 2021; Tian et al., 2022) design to leverage the vision-language model CLIP for the long-tailed classification. For example,

Figure 1: t-SNE (Van der Maaten & Hinton, 2008) visualization of textual features and visual features on fine-tuned CLIP, and refined visual features by our methods. We visualize 10 classes with the least number of image samples from Places-LT and each class has 80 text prompts by filling templates.

BALLAD (Ma et al., 2021) employs an auxiliary linear adapter and commonly-used re-sampling strategy to fine-tune CLIP adapt to the imbalanced image dataset. VL-LTR (Tian et al., 2022) designs a class-wise contrastive learning framework to fine-tune CLIP and a language-guided recognition head to classify the input images, using additional text data collected from Wikipedia. Although these methods have achieved the desired performance, they ignore the semantic complementarity between multi-modal data, specifically for such an imbalanced distribution. Therefore, designing an effective framework for using vision-language knowledge from CLIP under the circumstances of long-tailed distribution is still worth exploring.

To this end, we first explore how fine-tuned CLIP performs on the imbalanced image classification problem. We fine-tune the vision-language backbone of CLIP through contrastive learning on a specific long-tailed dataset. The textual and visual features extracted from the fine-tuned vision-language backbone are visualized in Figure 1, where the language inputs are built by filling prompt templates with text labels and the image inputs are from the highly imbalanced image dataset. We can observe that the textual features from fine-tuned CLIP are relatively more discriminative than the visual features, specifically for minority classes. Motivated by this observation, we aim to make full use of the advantage from textual features to help discriminative visual features learning and improve the long-tailed classification task. Additionally, we give more visualizations in App. A for further demonstration, where we find similar observations on majority classes.

As shown in the *Refined Visual Features* part of Figure 1, focusing on the fine-tuned textual features that are still nearly discriminative, we consider textual features as prototypes (denoted as the circle) for robust representation learning of entangled biased visual features (denoted as the triangle). After disentangled representation learning, visual features from the same class should be clustered together tightly around the corresponding prototypes, which will also be pushed far from each other. As a result, we can obtain a clear and discriminative representation class boundary for better recognition. Specifically, we achieve this goal by pushing the distribution of visual features toward their prototypes. We first fine-tune CLIP backbones on a target imbalanced dataset. Then we employ a linear adapter to refine the biased visual features. Considering the representation mismatch between refined visual features and fine-tuned textual features, we initialize a learnable linear classifier by text prompt embeddings for both keeping consistency and further classifying images. Due to the specific initialization, we regard the weights of the classifier as prototypes instead of using textual features directly. To measure the difference between distributions, we introduce a principled approach where we minimize the optimal transport (OT) distance (Peyré et al., 2019) to refine our biased visual features and continuously move prototypes toward class centers by updating models in unsupervised manner. For better use of supervised information to help balance vision representation, we design a supervised contrastive learning loss based on the optimal transport plan to achieve this goal. By introducing such a constraint, our method further effectively helps to entangle the coupled visual features. As a result, classes can obtain more discriminative feature distributions than before, thereby achieving better classification performance. Moreover, our framework can also be quickly combined with image-modality-based methods of addressing imbalanced vision recognition to achieve more performance improvements. We conduct extensive experiments on long-tailed vision recognition benchmarks and achieve significant improved performance, which shows the superiority of our method in using vision-language information to help imbalanced visual recognition. Besides, our method can easily expand to few-shot learning without significant modifications, demonstrating the well flexibility of our method.

## 2 Related work

**Long-Tailed Classification.** Researchers have proposed numerous methods for long-tailed classification tasks. One promising direction aims to adjust the original imbalanced data distribution through re-sampling ((Drummond & Holte, 2003; He & Garcia, 2009; Barandela et al., 2004; Van Hulse et al., 2007)) or data augmentation (Chu et al., 2020; Li et al., 2021; Park et al., 2022; Chou et al., 2020; Gao et al., 2023a; 2024). Besides, various re-weighting methods have been introduced by assigning greater weights to tail samples during optimization automatically or manually, which can be categorized into class-level manner (Lin et al., 2017; Cao et al., 2019; Cui et al., 2019; Li et al., 2025) and instance-level manner (Dong et al., 2017; Cui et al., 2019; Menon et al., 2020; Liu et al., 2022; Guo et al., 2022b). Some studies (Kang et al., 2019) have found that decoupling the feature extractor from the classifier can enhance the performance in long-tailed recognition tasks. Therefore, the followings (Zhong et al., 2021; Zhou et al., 2020b) explore more discriminative feature learning and balanced classifier optimization. Recently, several works focus on enhancing long-tailed vision recognition by using contrastive learning (Cui et al., 2021; Wang et al., 2021a; Suh et al.; Kang et al., 2021; Li et al.). PaCo designs a parametric contrastive learning loss to tackle long-tailed recognition (Cui et al., 2021).

**Vision-Language Model.** Recently, the vision-language models (VLMs) have experienced a revolution and made significant strides in various downstream vision tasks (Radford et al., 2021; Jia et al., 2021; Lu et al., 2019; Tan & Bansal, 2019; Chen et al., 2020; Ren et al., 2024). For example, CLIP (Radford et al., 2021) and ALIGN (Jia et al., 2021) learn powerful visual-textual representation via contrastive learning on large-scale image pairs, which achieve astonishing results on a wide spectrum of vision tasks without any fine-tuning. BLIP (Li et al., 2022a) introduces a novel VLM that transfers flexibly to both vision-language understanding and generation tasks.

Some works have been proposed to design vision-language approaches for long-tailed recognition. This paper (Wang et al., 2023) fixes the visual encoder of CLIP and employs a light decoder to adjust visual features first. Then they explore improvements of CLIP on imbalanced datasets using prompt tuning, fine-tuning and incorporating image-modality based methods. BALLAD (Ma et al., 2021) first fully fine-tunes CLIP on target imbalanced datasets and then refine the biased visual representation using a linear adapter with a balanced sampling strategy (Kang et al., 2019). Consequently, BALLAD classifies an input image by querying the maximum similarity with class text prompts in a CLIP manner. VL-LTR (Tian et al., 2022) presents a novel vision-language framework for imbalanced recognition. It collects additional text descriptions from Wikipedia and introduces a class-wise image-text contrastive loss for fully fine-tuning CLIP on target imbalanced datasets. Subsequently, VL-LTR employs a language-guided recognition head for classification by querying the maximum similarity between the input image and selected text prompts. While both BALLAD and VL-LTR outperform image-modality based methods on imbalanced recognition benchmarks, the learned visual representations remain heavily coupled due to the imbalanced data distribution.

## 3 background

**Long-Tailed Classification with Visual Modality.** Denote $\mathcal{D}_{\text{train}} = \{(\mathbf{x}_i, y_i)\}_{i=1}^{N}$ as the whole training data with length $N$, where $\mathbf{x}_i \in \mathbb{R}^d$ denotes the $i$-th image input with the dimension $d$ and $y_i$ indicates the corresponding numerical label over $K$ classes. Without loss of generality, we assume $N_1 > N_2 > ... > N_K$, where $N_k$ is the number of training samples in the class $k$. The prediction of a neural network usually consists of a feature extractor $f : \mathbf{x} \to \mathbf{u}$ and a linear classifier $g : \mathbf{u} \to \mathbf{y}$, where $\mathbf{u} \in \mathbb{R}^h$ means the $h$-dimensional representation. $f$ and $g$ can be parameterized with $\mathbf{W}_f$ and $\mathbf{W}_g$, respectively, which can be learned with empirical risk minimization over $\mathcal{D}_{\text{train}}$. However, this way ignores such class imbalance, resulting in poor performance on the minority classes.

**Optimal Transport.** OT has been widely used to measure the cost of transporting one probability measure to another. We focus on OT for discrete distributions and we refer the readers to Peyré et al. (2019) for more details. Assume we have two discrete distributions living in the arbitrary same space, formulated as $p = \Sigma_{i=1}^{n} a_i \delta_{x_i}$ and $q = \Sigma_{j=1}^{m} b_j \delta_{y_j}$. $\delta$ indicates the delta function, and both $\boldsymbol{a} \in \Delta^n$ and $\boldsymbol{b} \in \Delta^m$ denote the

probability simplex of $\mathbb{R}^n$ and $\mathbb{R}^m$, respectively. The OT distance between $p$ and $q$ can be expressed by:

$$\text{OT}(p, q) = \min_{\mathbf{T} \in \Pi(p,q)} \langle \mathbf{T}, \mathbf{C} \rangle = \sum_{i=1}^{n} \sum_{j=1}^{m} T_{ij} C_{ij}, \tag{1}$$

where $\mathbf{C} \in \mathbb{R}_{>0}^{n \times m}$ is the cost matrix whose element $C_{ij}$ denotes the distance between $x_i$ and $y_j$, and transport probability matrix $\mathbf{T} \in \mathbb{R}_{>0}^{n \times m}$ satisfies $\Pi(P, Q) := \left\{ \mathbf{T} \mid \sum_{i=1}^{n} T_{ij} = b_j, \sum_{j=1}^{m} T_{ij} = a_i \right\}$. An entropic constraint, $i.e., H(\mathbf{T}) = -\sum_{nm} T_{ij} \ln T_{ij}$, is introduced in the widely-used Sinkhorn algorithm (Cuturi, 2013) for discrete OT problems with reduced complexity.

**Revisit to CLIP.** CLIP (Radford et al., 2021) consists of a vision encoder $\mathcal{E}_{\text{vis}}$ and a language encoder $\mathcal{E}_{\text{lan}}$, which is trained to align the the visual and textual embedding spaces with a contrastive loss on a large training dataset consisting of 400 million image-text pairs. Specifically, for the $i$-th pair including the input image and the corresponding input text sequence, $\mathcal{E}_{\text{vis}}$ and $\mathcal{E}_{\text{lan}}$ are adopted to extract the visual feature $\mathbf{u}_i$ and textual feature $\mathbf{v}_i$, where $\mathbf{u}_i$ and $\mathbf{v}_i$ are both $h$-dimensional normalized vectors in the joint multimodal space. During pretraining, CLIP learn to align image-text pairs inside a batch of $B$ image-text pairs. The overall training objective consists of matching losses from two different directions, i.e., $\mathcal{L}_v$ for matching images to text and $\mathcal{L}_t$ for text-to-image matching. They both maximize the cosine similarity for matched pairs while minimize that of unmatched ones:

$$\mathcal{L}_v = -\frac{1}{B} \sum_{i=1}^{B} \frac{\exp((\mathbf{u}_i^\top \mathbf{v}_i)/\tau)}{\sum_{j=1}^{B} \exp((\mathbf{u}_i^\top \mathbf{v}_j)/\tau)}, \qquad \mathcal{L}_t = -\frac{1}{B} \sum_{i=1}^{B} \frac{\exp((\mathbf{v}_i^\top \mathbf{u}_i)/\tau)}{\sum_{j=1}^{B} \exp((\mathbf{v}_i^\top \mathbf{u}_j)/\tau)}, \tag{2}$$

where $\tau$ is a temperature hyperparameter, $(\mathbf{u}_i^\top \mathbf{v}_i)$ denotes the cosine similarity between $\mathbf{u}_i$ and $\mathbf{v}_i$, and $B$ is the batch size. Given the encoders are well pre-trained in massive image-text pairs, CLIP builds a strong connection between different modalities and thus shows the powerful capability of processing zero-shot visual recognition. Given $K$ candidate classes, we can construct a set of text prompts $\mathbf{t}$ with the length $K$ by filling description templates like `"a photo of a [label]."`, where the class token is replaced by the specific class name, such as "dog". Denote $\mathbf{u}$ be image feature extracted by the image encoder for the test image and $[\mathbf{v}_1, ..., \mathbf{v}_K]$ a set of weight vectors generated by the text encoder for the prompts $\mathbf{t}$. The classification probability of the test image towards class $k$ is computed as below:

$$p_k = \frac{\exp((\mathbf{u}^\top \mathbf{v}_k)/\tau)}{\sum_{j=1}^{K} \exp((\mathbf{u}^\top \mathbf{v}_j)/\tau)}. \tag{3}$$

Finally, the test image will be labeled to the class with the highest prediction probability.

**Fully Fine-tune CLIP.** To effectively leverage the target dataset, fully fine-tuning the textual and visual encoders in CLIP on $\mathcal{D}_{\text{train}}$ helps the model adapt to the downstream task. Simply, the text descriptions for images can be constructed by `"a photo of a [label]."` if the input data only includes images. Driven by the task, we can decide to utilize loss $\mathcal{L}_v$ or both of $\mathcal{L}_v$ and $\mathcal{L}_t$ in Eq. 2 for fine-tuning. After that, $\mathcal{E}_{\text{vis}}$ and $\mathcal{E}_{\text{lan}}$ can be adapted to the current dataset.

## 4 Methodology

### 4.1 Overall Framework

This paper proposes a novel framework based on the vision-language model to solve the long-tailed classification task, where we use CLIP as our backbone. As shown in Figure 2, our proposed framework includes two phases. Specifically, Phase A fully fine-tunes the CLIP model on the long-tailed training data following (Ma et al., 2021), which is mentioned in Section 3 and we only utilize loss $\mathcal{L}_v$ considering we focus on classifying the images correctly. However, recalling Figure 1, the fine-tuned textual features are more balanced and discriminative than the fine-tuned visual features. Below, we design Phase B which aims to learn discriminative visual features and a robust classifier, which is the focus of this work.

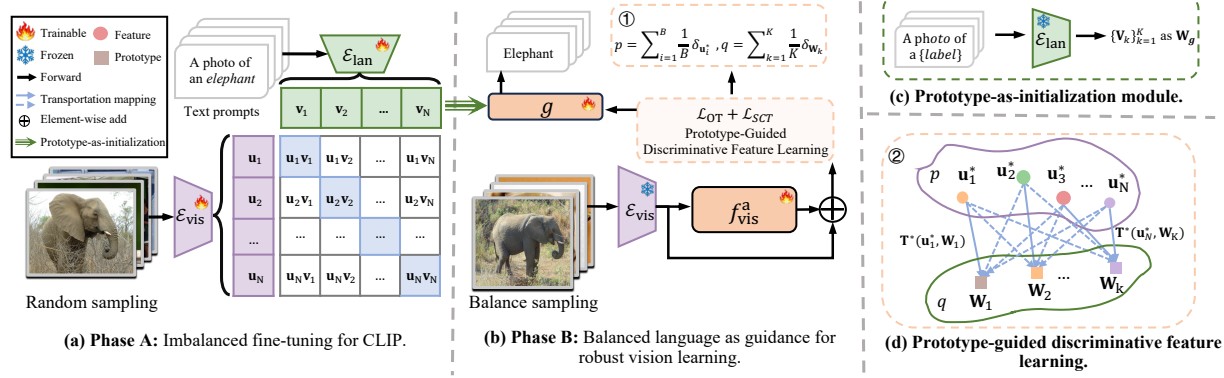

(a) **Phase A:** Imbalanced fine-tuning for CLIP.

(b) **Phase B:** Balanced language as guidance for robust vision learning.

(c) **Prototype-as-initialization module.**

(d) **Prototype-guided discriminative feature learning.**

Figure 2: **Training process of our proposed framework.** The overall method includes two phases. In Phase A, we fully fine-tune CLIP encoders randomly sampling examples from the imbalanced dataset. In Phase B, we employ a vision adapter $f_{\text{vis}}^{\text{a}}$ to refine biased visual features from $\mathcal{E}_{\text{vis}}$ and a classifier $g$ for classification, where $g$ will be initialized by textual features using $\mathbf{v}$ from $\mathcal{E}_{\text{lan}}$ and we adopt a balanced sampling. We then build vision representation distribution $p$ by refined visual features $\mathbf{u}^*$ and prototype distribution $q$ by the weights $\mathbf{W}$ of $g$. We finally learn discriminative visual features and robust classifier by OT-based prototype-guided mapping between $p$ and $q$.

## 4.2 Phase B: Visual representation refining and prompt-oriented classifier training

**Linear adapter for refining visual features**. As shown in part (b) in Figure 2, different from Phase A, we adopt a class-balanced sampling strategy (Kang et al., 2019) to train the concerned model, which constructs a balanced group of training samples. Inspired by parameter-efficient adapter modules (Ma et al., 2021; Gao et al., 2023b; Houlsby et al., 2019), we introduce an additional linear adapter layer $f_{\text{vision}}^{\text{adapter}} \in \mathbb{R}^{h \times h}$ ($f_{\text{vis}}^{\text{a}}$ short) to refine the visual feature $\mathbf{u}$, where we freeze the visual-language backbone obtained from Phase A. Specifically, given the image-text pair, the textual feature $\mathbf{v}$ keep unchanged in Phase A while the visual feature is further refined as follows:

$$\mathbf{u}^* = \beta \cdot f_{\text{vis}}^{\text{a}}(\mathbf{u}) + (1 - \beta) \cdot \mathbf{u}, \tag{4}$$

where $\beta$ is a residual factor for combining the visual feature $f_{\text{vis}}^{\text{a}}(\mathbf{u})$ and the visual feature $\mathbf{u}$ from the from the frozen visual encoder $\mathcal{E}_{\text{vis}}$ in Phase A.

A straightforward approach to optimize the linear adapter is using $\mathcal{L}_v$ in Eq. 2 by replacing $\mathbf{u}$ with $\mathbf{u}^*$ directly, similar to BALLAD (Ma et al., 2021), where the frozen textual features are viewed as a fixed classifier. However, this approach ignores the mismatch between refined visual features $\mathbf{u}^*$ from the learnable adapter and textual features $\mathbf{v}$ from the frozen textual encoder.

**Learnable classifier initialized by text embeddings.** To address the mismatch issue between refined visual features and textual features and make full use of the valuable knowledge embedded within the text modality, we introduce a learnable linear classifier and initialize it with textual features associated with the class labels. As shown in part (c) in Figure 2, denote the linear classifier $g : \mathbb{R}^h \to \mathbb{R}^K$ is parameterized by $\mathbf{W}_g \in \mathbb{R}^{h \times K}$ for ignoring the bias term, where $\mathbf{W}_g = \{\boldsymbol{w}_1, ..., \boldsymbol{w}_K\}$. We initialize the classifier weight $\boldsymbol{w}_k$ using the textual feature $\mathbf{v}_k$ from fine-tuned $\mathcal{E}_{\text{lan}}$ with $k = 1 : K$. After we obtain the semantic information initialized classifier, we can optimize the linear adapter $f_{\text{vis}}^{\text{a}}$ and classifier $g$ using cross-entropy (CE) loss as follows:

$$\mathcal{L}_{\text{CE}} = -\frac{1}{B} \sum_{i=1}^{B} \sum_{k=1}^{K} y_{i,k} \cdot \log p_k = -\frac{1}{B} \sum_{i=1}^{B} \sum_{k=1}^{K} y_{i,k} \cdot \log \frac{\exp(g(\mathbf{u}_i^*)_k)/\tau}{\sum_{j=1}^{K} \exp(g(\mathbf{u}_i^*)_j)/\tau}, \tag{5}$$

where $y_{i,k} = \{0, 1\}$ and $y_{i,k} = 1$ only if the input image $x_i$ belongs to class $k$. $B$ is the batch size. Moving beyond BALLAD, which uses the text encoder from Phase A as the fixed classifier and learns the adapter with the contrastive loss, we provide a learnable classifier initialized by the text embeddings and optimize it with the CE loss. Therefore, our proposed approach can not only avoid the mismatch problem between refined features and textual features but also leverage rich semantic information within the text modality. In addition, existing long-tailed classification methods based on the image modality usually adopt CE loss,

---

**Algorithm 1:** Training process.

**Input** : Dataset $\mathcal{D}_{\text{train}}$, encoders $\mathcal{E}_{\text{vis}}$ and $\mathcal{E}_{\text{lan}}$, adapter $f_{\text{vis}}^{\text{a}}$ and classifier $g$, hyper-parameters:$\{\lambda_1, \lambda_2, \gamma, \beta$ and $\tau\}$

**1** Initialize $\mathcal{E}_{\text{vis}}$ and $\mathcal{E}_{\text{lan}}$ with CLIP;

**2** **for** $epoch = 1, ..., E_1$ **do**                           `// Phase A. Fine-tuning backbones`

**3**      **for** *Random sample a mini-batch* $\{(\mathbf{x}, \mathbf{y})\}_{n=1}^{B} \sim \mathcal{D}_{train}$ **do**

**4**          Obtain the visual features $\mathbf{u} = \mathcal{E}_{\text{vis}}(\mathbf{x})$;

**5**          Obtain the text prompts $\mathbf{t}$ by tokenization $\mathbf{t} = tokenize(\mathbf{y})$;

**6**          Obtain the textual features $\mathbf{v} = \mathcal{E}_{\text{lan}}(\mathbf{t})$;

**7**          Update $\mathcal{E}_{\text{vis}}$ and $\mathcal{E}_{\text{lan}}$ by minimizing $\mathcal{L}_{\text{A}} = \mathcal{L}_v = -\frac{1}{B} \sum_{i=1}^{B} \frac{\exp((\mathbf{u}_i^\top \mathbf{v}_i)/\tau)}{\sum_{j=1}^{B} \exp((\mathbf{u}_i^\top \mathbf{v}_j)/\tau)}$;

**8**      **end**

**9** **end**

**10** Freeze $\mathcal{E}_{\text{vis}}$ and $\mathcal{E}_{\text{lan}}$;

**11** Random initialize the vision adapter $f_{\text{vis}}^{\text{a}}$;

**12** Initialize the classifier $g$ by setting $\mathbf{W}_g = \mathcal{E}_{\text{lan}}(\mathbf{t})$;

**13** **for** $epoch = E_1 + 1, ..., E_1 + E_2$ **do**             `// Phase B. Updating the adapter and classifier.`

**14**      **for** *Balance sample a* $\{(\mathbf{x}, \mathbf{y})\}_{n=1}^{B} \sim \mathcal{D}_{train}$ **do**

**15**          Obtain the refined visual features $\mathbf{u}^* = \beta f_{\text{vis}}^{\text{a}}(\mathcal{E}_{\text{vis}}(\mathbf{x})) + (1 - \beta)\mathcal{E}_{\text{vis}}(\mathbf{x})$ ;

**16**          Use $\mathbf{u}^*$ and the weights $\mathbf{W}_g$ of $g$ to build $p$ and $q$, see Section 22;

**17**          Calculate the classification loss $\mathcal{L}_{\text{CE}} = -\frac{1}{B} \sum_{i=1}^{B} \sum_{j=1}^{K} y_{i,j} \cdot \log \frac{\exp(g(\mathbf{u}_i^*)_j)/\tau)}{\sum_{k=1}^{K} \exp(g(\mathbf{u}_i^*)_k)/\tau)}$;

**18**          Obtain the optimal transport plan $\mathbf{T}^*$ by Eq. 8 and calculate OT distance $\mathcal{L}_{\text{OT}}(p, q)$ by Eq. 5;

**19**          Given $\mathbf{T}^*$, calculate the supervised contrastive distance $\mathcal{L}_{\text{SCT}} = \frac{1}{B} \sum_{i=1}^{B} \frac{\frac{1}{|I_i^+|} \sum_{j \in I_i^+} ||\mathbf{T}_i^* - \mathbf{T}_j^*||_1}{\frac{1}{|I_i^-|} \sum_{k \in I_i^-} ||\mathbf{T}_i^* - \mathbf{T}_k^*||_1}$;

**20**          Update $f_{\text{vis}}^{\text{a}}$ and $g$ by minimizing $\mathcal{L}_{\text{B}} = \mathcal{L}_{\text{CE}} + \lambda_1 \mathcal{L}_{\text{OT}} + \lambda_2 \mathcal{L}_{\text{SCT}}$;

**21**      **end**

**22** **end**

---

such as re-weighting method (Cui et al., 2019; Kang et al., 2019; Zhong et al., 2021), making it possible for combining our approach with these existing long-tailed methods.

**Unsupervised prototype-guided feature learning by Optimal Transport.** Although the linear adapter and classifier can be optimized by the CE loss, we can further provide an unsupervised loss function to improve their learning quality from the view of distribution matching. More specifically, Figure 1 reveals that textual embeddings are more discriminative and balanced than visual embeddings and we initialize the classifier with the textual embeddings, where we consider the classifier weights as the prototypes. Therefore, it is reasonable to align the distribution of the prototypes from the textual embeddings with the distribution of the refined visual representations in an unsupervised manner, where we use the optimal transport. As shown in part (d) in Figure 2, taking a mini-batch $B$ of refined visual features as the example, we can denote its empirical distribution as $p = \sum_{i=1}^{B} \frac{1}{B} \delta_{\mathbf{u}_i^*}$. Besides, the empirical distribution of $K$ prototypes from all classes can be formulated as $q = \sum_{k=1}^{K} \frac{1}{K} \delta_{\boldsymbol{w}_k}$. In order to align the distributions of prototypes and refined visual features, we can minimize the OT distance between $p$ and $q$, formulated as follows:

$$\mathcal{L}_{\text{OT}}(p, q) = \min_{\mathbf{T} \in \Pi(p,q)} \langle \mathbf{T}, \mathbf{C} \rangle - \gamma H(\mathbf{T}), \tag{6}$$

where $\gamma > 0$ is a hyper-parameter for the entropy constraint $H(\mathbf{T})$, the transport plan $\mathbf{T}$ satisfies $\Pi(p, q) := \left\{ \mathbf{T} | \sum_{i=1}^{B} \mathbf{T}_{nk} = 1/K, \sum_{k=1}^{K} \mathbf{T}_{ik} = 1/B \right\}$, and the cost function $\mathbf{C}_{ik}$ measures the distance between visual feature $\mathbf{u}_i^*$ and a prototype $\boldsymbol{w}_k$. Without specific instructions, we use cosine similarity as the distance metric, *i.e.*, $\mathbf{C}_{ik} = 1 - \cos(\mathbf{u}_i^*, \boldsymbol{w}_k)$, though other reasonable choices can also be used here. Intuitively, minimizing this expected moving cost encourages the refined visual features and prototypes to be aligned. Ideally, a refined visual feature should be close to its corresponding class prototype and far away from other class

prototypes according to their similarity. Therefore, introducing the OT loss is helpful for learning more discriminative visual features and robust classifier.

**Contrastive supervised constraint on transport plan.** Notably, the introduced OT loss serves as an unsupervised regularization term of the CE loss. We can further introduce a supervised contrastive constraint based on the learned transport plan to assist the learning of the adapter encoder and classifier weights (prototypes). The motivation of this constraint is that the samples from the same class should close to each other while the samples from different classes should be far apart, where the label of input image is available. Considering the learned optimal transport plan $\mathbf{T}^*$ depends on the cost matrix $\mathbf{C}$, which relies on the to-be-learned linear adapter and prototypes, we can view the $i$-th row of transport plan matrix $\mathbf{T}^*$ as another feature of the $i$-th sample. Now, the similarity between $\mathbf{T}_i^*$ and $\mathbf{T}_j^*$ can inherit the relationship between the $i$-th sample and the $j$-th sample. Therefore, we formulate the supervised contrastive loss as follows:

$$\mathcal{L}_{\text{SCT}} = \frac{1}{B} \sum_{i=1}^{B} \frac{\frac{1}{|I_i^+|} \sum_{j \in I_i^+} ||\mathbf{T}_i^* - \mathbf{T}_j^*||_1}{\frac{1}{|I_i^-|} \sum_{k \in I_i^-} ||\mathbf{T}_i^* - \mathbf{T}_k^*||_1}, \tag{7}$$

where $|| \cdot ||_1$ indicates the L1 distance, $I_i^+$ denotes a positive set (has the same label with sample $i$) of $\mathbf{T}_i^*$ and $I_i^-$ is a negative set. Designed in this way the refined visual features from the same class are optimized to be close to each other; and vice versa, improving the intra-class compactness and inter-class separability. We can obtain the optimal transport plan $\mathbf{T}^* \in \mathbb{R}_{>0}^{N \times K}$ with a fast optimization solution in a few iterations:

$$\mathbf{T}^* = \text{diag}(\mathbf{a}^{(t)})\exp(-\mathbf{C}/\gamma)\text{diag}(\mathbf{b}^{(t)}), \tag{8}$$

where $t$ indicates the current iteration and in each iteration, $\mathbf{a}^{(t)} = \mathbf{a}/((\exp(-\mathbf{C}/\gamma)\mathbf{b}^{(t-1)}))$ and $\mathbf{b}^{(t)} = \mathbf{b}/((\exp(-\mathbf{C}/\gamma)^\top \mathbf{a}^{(t-1)}))$ with $\mathbf{a}^{(0)} = \frac{1}{\mathbf{B}}$ and $\mathbf{b}^{(0)} = \frac{1}{\mathbf{K}}$.

**Overall loss function for Phase B.** In Phase B, the overall objective function for optimizing the vision adapter $f_{\text{vis}}^{\text{a}}$ and classifier $g$ is

$$\mathcal{L}_{\text{B}} = \mathcal{L}_{\text{CE}} + \lambda_1 \mathcal{L}_{\text{OT}} + \lambda_2 \mathcal{L}_{\text{SCT}}, \tag{9}$$

where $\lambda_1$ and $\lambda_2$ are hyper-parameters. We give an overall algorithm process of our method shown in 1.

In conclusion, based on the observation that fine-tuned visual features from CLIP are more entangled than textual features, we design to employ a linear adapter to refine the coupled vision representation and a learnable classifier initialized by text embeddings to keep the consistency between image and text modality. Further, by considering the weights of the classifier as prototypes, we design an unsupervised prototype-guided feature learning loss to help separate vision distribution by optimal transport. To better use the supervised signals, we propose a contrastive supervised loss on transport plan to further robust training and classification. Our proposed method gives a novel and effective framework motivated by the natural discrepancy between data distributions to address the issue of long-tailed vision recognition based on the vision-language model.

## 5  Experiments

We conduct extensive experiments to evaluate our method, including comparison with SOTA methods on benchmarks, ablation studies, visualization, etc. The imbalance factor (IF) of a dataset can be defined as the data point amount ratio between the largest and smallest classes.

**Datasets and Evaluation Metric.** We evaluate our methods on ImageNet-LT (Liu et al., 2019), Places-LT (Liu et al., 2019) and iNaturalist2018 (Van Horn et al., 2018). Among these datasets, ImageNet-LT is a subset from ImageNet-2012 (Deng et al., 2009) by downsampling, which contains 115.8K training images from 1000 classes with IF $= \frac{1280}{5}$. Places-LT is also a long-tailed version of Placse-365 dataset (Zhou et al., 2017), which includes 62.5K images from 365 classes and IF $= \frac{4980}{5}$. iNaturalist2018 dataset (Van Horn et al., 2018) is a real-world dataset for fine-grained and long-tailed classification tasks, which has 437.5K training images with IF $= \frac{1000}{2}$. We evaluate the method on the corresponding test datasets, which are all balanced for 50 images per class in ImageNet-LT, 100 for Places-LT and 3 for iNaturalist2018. Following the

Table 1: Test top-1 accuracy (%) on ImageNet-LT and Places-LT. All the backbones are initialized with CLIP weights and based on the ResNet and ViT structure.†, ‡ and * indicates the results from Ma et al. (2021), Tian et al. (2022) and Zhou et al. (2022), respectively. All the backbones are initialized with CLIP weights. -a indicates using additional texts for training.

| Method | Backbone | ImageNet-LT | | | | Places-LT | | | |
|---|---|---|---|---|---|---|---|---|---|
| | | Many | Medium | Few | Overall | Many | Medium | Few | Overall |
| NCM (Kang et al., 2019) | RN50‡ | 58.9 | 46.6 | 31.1 | 49.2 | 37.1 | 30.6 | 19.9 | 30.8 |
| cRT (Kang et al., 2019) | RN50‡ | 63.3 | 47.2 | 27.8 | 50.8 | 38.5 | 29.7 | 17.6 | 30.5 |
| LWS (Kang et al., 2019) | RN50‡ | 62.2 | 48.6 | 31.8 | 51.5 | 36.0 | 32.1 | 20.7 | 31.3 |
| $\tau$-normalized (Kang et al., 2019) | RN50‡ | 60.9 | 48.4 | 33.8 | 51.2 | 34.5 | 31.4 | 23.6 | 31.0 |
| PaCo (Cui et al., 2021) | RN50† | - | - | - | 60.2 | - | - | - | - |
| Zero-Shot CLIP (Radford et al., 2021) | RN50‡ | 60.8 | 59.3 | 58.6 | 59.8 | 37.5 | 37.5 | 40.1 | 38.0 |
| BALLAD (Ma et al., 2021) | RN50† | 71.0 | 66.3 | 59.5 | 67.2 | 46.7 | 48.0 | 42.7 | 46.5 |
| VL-LTR-a (Tian et al., 2022) | RN50‡ | 77.8 | 67.0 | 50.8 | 70.1 | 51.9 | 47.2 | 38.4 | 48.0 |
| VL-LTR (Tian et al., 2022) | RN50‡ | 77.9 | 66.5 | 49.3 | 69.4 | 52.7 | 46.8 | 36.3 | 47.3 |
| OURS | RN50 | 72.1 | 70.9 | 63.4 | **70.3** | 47.5 | 49.5 | 50.5 | **49.0** |
| Method | Backbone | ImageNet-LT | | | | Places-LT | | | |
| | | Many | Medium | Few | Overall | Many | Medium | Few | Overall |
| Zero-Shot CLIP (Radford et al., 2021) | ViT-B/16 | 69.2 | 67.6 | 67.7 | 68.3 | 38.3 | 39.2 | 45.9 | 40.2 |
| CLIP+LP (Radford et al., 2021) | ViT-L/14* | 87.3 | 65.0 | 18.9 | 67.4 | 55.6 | 34.4 | 14.4 | 38.2 |
| CoOp (Zhou et al., 2022) | ViT-L/14* | - | - | - | 60.3 | - | - | - | 26.1 |
| BALLAD (Ma et al., 2021) | ViT-B/16† | 79.1 | 74.5 | 69.8 | 75.7 | 49.3 | 50.2 | 48.4 | 49.5 |
| VL-LTR (Tian et al., 2022) | ViT-B/16‡ | 84.5 | 74.6 | 59.3 | 77.2 | 54.2 | 48.5 | 42.0 | 50.1 |
| OURS | ViT-B/16 | 81.4 | 77.7 | 71.7 | **78.2** | 49.8 | 52.6 | 54.2 | **51.9** |

common setting(Liu et al., 2019; Kang et al., 2019; Ma et al., 2021; Tian et al., 2022), we report the overall top-1 accuracy and also report the top-1 accuracy of many-shot ($\geq 100$ samples), medium-shot ($100 \sim 20$ samples) and few-shot ($\leq 20$ samples) according to the number of training samples in each class.

**Baselines.** We can compare our framework with the following methods: 1) **Visual encoder of CLIP + imbalanced learning algorithms**: The classical imbalanced learning methods designed for the image-modality are re-implemented using the pre-trained visual encoder of CLIP by previous works (Tian et al., 2022; Ma et al., 2021), including NCM (Kang et al., 2019), cRT (Kang et al., 2019), LWS (Kang et al., 2019), $\tau$-noimalized (Kang et al., 2019) and PaCo (Cui et al., 2021). PaCo falls into contrastive learning and others belong to the decoupling method. 2) **CLIP and its variants**: Considering we utilize CLIP as our backbone, we select zero-shot CLIP, CLIP + linear probing (Radford et al., 2021) and CoOp (Zhou et al., 2022) as baselines. 3) **Multi-Modality Methods**: Methods employ CLIP model and text prompts for imbalanced vision recognition, including BALLAD (Ma et al., 2021) and VL-LTR (Tian et al., 2022), which are our main competitors. Besides, we also comprehensively compare our method with competitive image-modality-based methods in App. C.

**Implementation details.** We employ ResNet-50 (He et al., 2016) or ViT-B/16 (Dosovitskiy et al., 2020) as our visual backbone, and a 12-layer transformer (Radford et al., 2019) as the language backbone, where backbones are initialized by CLIP weights and detailed training hyper-parameters can be found in App. B.

## 5.1 Experiments on ImageNet-LT and Places-LT

**Results.** As listed in Table 1, our method outperforms the baseline methods by a significant margin on both datasets, especially on the few-shot setting. In particular, our method achieves 70.3% and 49.0% overall accuracy on ImageNet-LT and Places-LT when equipped with ResNet-50, which is 3.1% and 2.5% higher than BALLAD, and 0.2% and 1.0% higher than VL-LTR. Based on ViT-B/16, our method achieves 78.2% and 51.9% overall accuracy, which is 2.5% and 2.4% higher than BALLAD, and 1.0% and 1.8% higher than VL-LTR. This indicates our method outperforms the previous under different backbones. Besides, for BALLAD, our method has a comprehensive lead in terms of the accuracy on three different shots. For VL-LTR, our method especially obtains large performance gains on concerned tail classes. We can observe that our method has 12.6% and 12.1% improvements on the few-shot split on ResNet-50 for both two datasets, and that is 12.4% and 12.2% on ViT/B-16, when compared with VL-LTR (w/ additional texts). Even for medium-shot split, our method also has a significant performance improvement over VL-LTR. On the other

hand, our method does not require additional text descriptions and still outperforms VL-LTR with less training data. When using the same amount of training data (prompt templates only, no additional text descriptions), our method shows a more significant performance improvement over VL-LTR from 0.2% to 0.9% on ImageNet-LT and 1.0% to 1.7% on Places-LT based on ResNet-50. These results suggest that our method is a more effective approach for long-tailed image classification based on CLIP.

## 5.2 Experiments on iNaturalist2018

**Results.** As shown in Table 2, we can see that our proposed method achieves better performance on iNaturalist2018 under both ResNet-50 and ViT-B/16. Compared to zero-shot CLIP, VL-LTR and our method both achieve significant performance improvements, which are mainly due to the better utilization of CLIP for imbalanced fine-grained vision classification. On ResNet-50, our method outperforms BALLAD by 1.3% and VL-LTR by 0.9%. On ViT-B/16, we have a 0.8% improvement than VL-LTR when the input resolution is $3 \times 224 \times 224$, and it is 1.3% when the resolution is $3 \times 384 \times 384$. Our experiments show that our method can achieve higher performance across different backbones, especially without using any additional text data for training.

## 5.3 Ablation Study

In this section, we conduct extensive ablation studies to investigate reasons for the performance improvement of our method. Without being specifically mentioned, the training settings remain the same as in Section 5.1 and Section 5.2.

**Proposed Loss.** To examine the effectiveness of the proposed loss functions, we perform comprehensive ablation studies on loss functions, where we report top-1 overall accuracy. ✓ indicates its coefficient of 0.1 and - represents that of 0. As shown in Table 3, with the help of $\mathcal{L}_{\mathrm{OT}}$ and $\mathcal{L}_{\mathrm{SCT}}$, performance on all three datasets increases 0.55%, 0.66% and 0.28% on ResNet-50 and 0.45%, 0.43% and 0.46% on ViT-B/16. These results prove the effectiveness and generality of the proposed loss functions in helping improve performance. Beyond the accuracy, we also observe that the introduction of $\mathcal{L}_{\mathrm{OT}}$ and $\mathcal{L}_{\mathrm{SCT}}$ helps convergence of the optimization as shown in Figure 3.

Table 2: Test top-1 accuracy (%) on iNaturalist2018. All the backbones are initialized with CLIP weights except for PaCo. †, ‡ and * indicates the results from (Ma et al., 2021), (Tian et al., 2022) and (Wang et al., 2023). All the backbones are initialized with CLIP weights. - $a$ indicates using additional texts for training.

| Method | Backbone | iNaturalist2018 Overall |
|---|---|---|
| NCM (Kang et al., 2019) | RN50‡ | 65.3 |
| cRT (Kang et al., 2019) | RN50‡ | 69.9 |
| LWS (Kang et al., 2019) | RN50‡ | 71.0 |
| $\tau$-normalized (Kang et al., 2019) | RN50‡ | 71.2 |
| PaCo (Cui et al., 2021) | RN50† | 73.8 |
| Zero-shot CLIP (Radford et al., 2021) | RN50‡ ViT-L/14* | 3.4 5.5 |
| BALLAD (Ma et al., 2021) | RN50† | 74.2 |
| VL-LTR-a (Tian et al., 2022) | RN50‡ ViT-B/16‡ | 74.6 76.8 |
| OURS | RN50 ViT-B/16 | **75.5** **77.6** |
| VL-LTR-a-384 | ViT-B/16‡ | 81.0 |
| OURS-384 | ViT-B/16 | **82.3** |

Table 4: Ablation on architecture on ImageNet-LT and Places-LT with ViT-B/16 as the backbone.

Table 3: Ablation on proposed loss functions on ImageNet-LT, Places-LT and iNaturalist2018 (iNat2018) with different backbones.

| Backbone | $\mathcal{L}_{\mathrm{CE}}$ | $\mathcal{L}_{\mathrm{OT}}$ | $\mathcal{L}_{\mathrm{SCT}}$ | ImageNet-LT | Places-LT | iNat2018 |
|---|---|---|---|---|---|---|
| RN50 | ✓ | - | - | 69.37 | 48.33 | 74.93 |
| | ✓ | ✓ | - | 70.07 | 48.71 | - |
| | ✓ | ✓ | ✓ | **70.27** | **49.03** | **75.21** |
| ViT-B/16 | ✓ | - | - | 77.10 | 50.89 | 77.09 |
| | ✓ | ✓ | - | 77.84 | 51.64 | - |
| | ✓ | ✓ | ✓ | **78.25** | **51.89** | **77.55** |

| $f_{\mathrm{vis}}^{\mathrm{a}}$ | $f_{\mathrm{lan}}^{\mathrm{a}}$ | $g$ | ImageNet-LT | Places-LT |
|---|---|---|---|---|
| - | - | - | 72.62 | 40.83 |
| ✓ | - | - | **76.79** | **49.87** |
| - | ✓ | - | 73.22 | 49.34 |
| ✓ | ✓ | - | 76.49 | 49.50 |
| - | - | ✓ | 77.52 | 51.44 |
| ✓ | - | ✓ | **78.24** | **51.89** |
| ✓ | ✓ | ✓ | 77.36 | 51.14 |

**Architecture.** To explore how the proposed architecture influences performance, we provide ablation studies on the architecture. We also take the language adapter for comparison, even we though do not adopt it in our framework shown in Table 4. Overall, $f_{\mathrm{vis}}^{\mathrm{a}}$, $f_{\mathrm{lan}}^{\mathrm{a}}$ and $g$ indicate the vision adapter, language adapter and classifier, respectively. The first group represents the framework without classifier and thus we only use $\mathcal{L}_{\mathrm{CE}}$ as the optimization objective to update adapters. As can be seen, the best choice is to only apply the linear adapter to the visual branch. The second group represents the framework with the linear classifier $g$

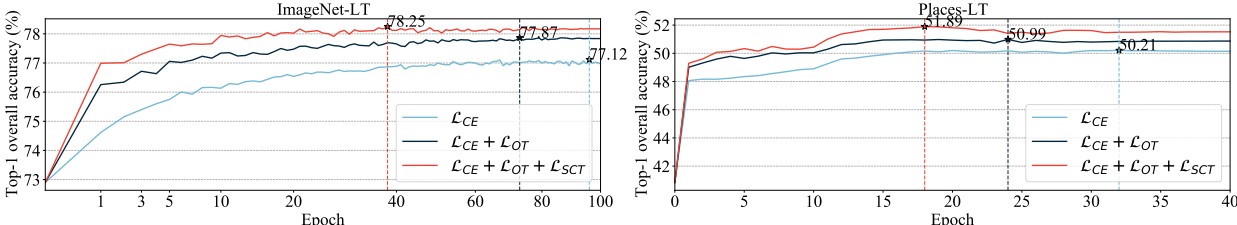

Figure 3: Visualization of top-1 overall accuracy per training epoch on ImageNet-LT and Places-LT on ViT-B/16.

Table 5: Experiments on the combination of our framework with other methods on ViT-B/16.

| Method | ImageNet-LT | | | | Places-LT | | | |
|---|---|---|---|---|---|---|---|---|
| | Many | Medium | Few | Overall | Many | Medium | Few | Overall |
| BALLAD (Ma et al., 2021) | 79.1 | 74.5 | 69.8 | 75.7 | 49.3 | 50.2 | 48.4 | 49.5 |
| VL-LTR (Tian et al., 2022) | 84.5 | 74.6 | 59.3 | 77.2 | 54.2 | 48.5 | 42.0 | 50.1 |
| OURS | 81.42 | 77.73 | 71.72 | 78.24 | 49.82 | 52.56 | 54.20 | 51.89 |
| CB-CE (Cui et al., 2019) | 81.18 | 77.73 | 72.14 | 78.29$_{\uparrow 0.04}$ | 50.28 | 52.24 | 54.25 | 51.99$_{\uparrow 0.10}$ |
| +LWS (Kang et al., 2019) | 81.18 | 77.74 | 72.17 | 78.30$_{\uparrow 0.05}$ | 50.18 | 52.24 | 54.24 | 52.02$_{\uparrow 0.13}$ |
| +MisLAS (Zhong et al., 2021) | 81.03 | 77.85 | 72.44 | 78.34$_{\uparrow 0.10}$ | 50.22 | 52.32 | 54.72 | 52.07$_{\uparrow 0.18}$ |
| +TTA (Shanmugam et al., 2021) | 81.29 | 78.39 | 73.12 | 78.79$_{\uparrow 0.55}$ | 50.24 | 53.04 | 54.76 | 52.42$_{\uparrow 0.53}$ |

initialized by text embeddings and we use $\mathcal{L}_\text{B}$ to optimize models. A comparison of the second and fourth rows shows that our approach already achieves better classification performance when combined with the classifier $g$ alone, with improvements of 0.73% and 1.57%. Furthermore, in the sixth row, the combination of $f_\text{vis}^\text{a}$ and $g$ shows the optimal choice for both datasets.

**Combination with other methods for further performance improvement.** Due to employing a linear classifier for vision recognition, we can leverage classifier-based strategies for vision-modality imbalanced methods into our framework for further improvement. In order to demonstrate the generality and the effectiveness of our framework, we conduct experiments on the combination of our method with loss re-weighting (class balanced loss (Cui et al., 2019)), decoupling (LWS (Kang et al., 2019), MisLAS (Zhong et al., 2021)) and test-time aggregation (TTA (Shanmugam et al., 2021)). As shown in Table 5, we can observe that with the help of TTA, our method obtains the most performance improvement by 0.55% on ImageNet-LT and 0.53% on Places-LT. The results show the generality of our proposed framework in addressing the long-tailed recognition issue by cooperating with other methods.

**Visualization of logits.** As shown in Figure 4, we visualize output logits from the classifier $g$ on ImageNet-LT based on ViT-B/16 using t-SNE (Van der Maaten & Hinton, 2008), where we randomly select the test data of 10 classes. In terms of CE loss as the optimization objective, we can observe that the output logits are entangled heavily. As expected, our method helps logits decoupled by making visual features more discriminative and prototypes far from each other. Therefore, it reveals visually balanced textual features as prototypes can help tackle imbalanced vision recognition.

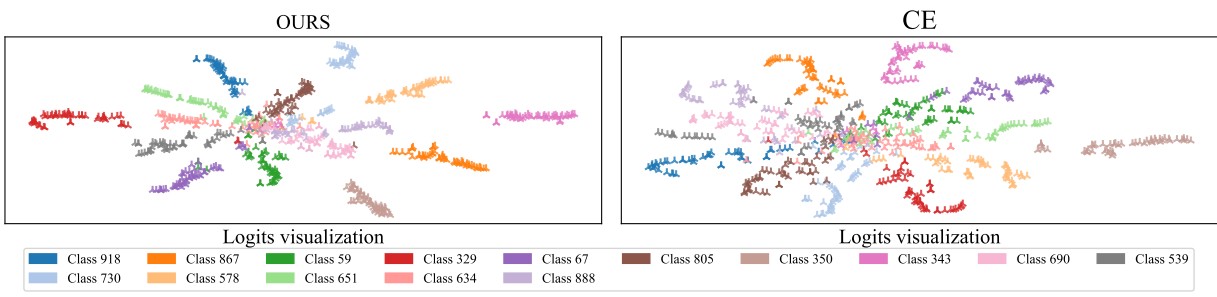

Figure 4: Visualization of the output logtis on ImageNet-LT on ViT-B/16.

### 5.4  Learnable Linear Classifier vs Frozen Linear Classifier

Given that we utilize textual prototypes as the initialization for the linear classifier and subsequently introduce an additional loss to iteratively update the classifier, there exists a potential risk of diminishing the consistency between visual features and text prototypes (classifier weights). This concern is examined through both experimental results and visualizations.

Firstly, as show in Table 6, results indicate a noteworthy performance improvement facilitated by the learnable linear classifier. This suggests that, contrary to the potential risk, the introduced update mechanism actually reinforces the consistency between visual features and text prototypes. We posit that the weights of the classifier serve as the evolved textual prototypes, and the observed performance enhancement implies that these weights have undergone substantial learning. This, in turn, signifies an improvement in the textual prototypes and a heightened alignment and coherence with visual features.

Table 6: Ablation on the update of the linear classifier on ViT/B-16.

| Batch Size | Places-LT | | | |
|---|---|---|---|---|
| | Many | Medium | Few | Overall |
| Learnable classifier | 49.8 | 52.6 | 54.2 | 51.9 |
| Frozen classifier | 49.0 | 51.4 | 50.8 | 50.4 |

On the other hand, we provide visualizations of refined visual features and their corresponding textual prototypes (classifier parameters) under scenarios where the classifier is frozen versus when it is learnable. As shown in Figure 5, the visual results demonstrate that the learnable textual prototypes (linear classifier) exhibit a more effective convergence toward the center of visual features. This observation attests to the well-maintained consistency between the two components. Moreover, the preserved consistency, in turn, facilitates a tighter clustering of the learned visual features around their respective prototypes. Therefore, from a visual standpoint, this underscores the indispensability of the learnable linear classifier. The visual evidence affirms that the learnable linear classifier is not only beneficial but also crucial in sustaining the alignment between visual features and textual prototypes, ultimately leading to a more balanced representation.

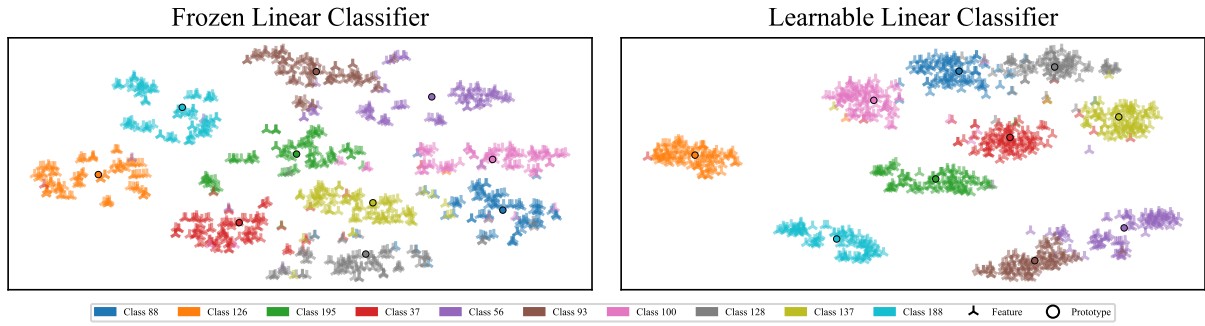

Figure 5: visualizations of refined visual features and their corresponding textual prototypes (classifier parameters) under scenarios where the classifier is frozen versus when it is learnable.

### 5.5  The effectiveness of the utilized decoupling training scheme

As mentioned in Section 4, we decouple the model training into two phases. In Phase A, we fine-tune the vision and language encoders of pre-trained CLIP on a target imbalanced dataset. In Phase B, we utilize a linear adapter for further discriminate vision representation learning and a classifier for robust long-tailed recognition. Another learning scheme for the model is to jointly train two encoders, the adapter and classifier rather than decoupling the training process. As shown in Table. 8, we can first observe that our method based on the fine-tuned CLIP gives the best performance. When compared with the joint training scheme for training two phases at the same time, the decoupling training scheme shows better performance in helping separate vision representation learning. This demonstrates that our proposed decoupled training of the model is more effective in handling imbalanced distribution.

## 5.6 Expansion to Few-shot Learning

Our motivation, to guide the learning of robust visual features through the use of textual representation, also holds the potential for enhancing visual reasoning capabilities in scenarios with limited data, for example, the few-shot learning. Here, we conduct experiments on the Caltech101 dataset, combining our method with Tip-Adapter (Zhang et al., 2021) and PLOT (Chen et al., 2022). In the implementation, we utilize textual features to initialize the classifier and then apply our proposed $\mathcal{L}_{\mathrm{OT}}$ and $\mathcal{L}_{\mathrm{SCT}}$. As shown in Table 7, results indicate that our method continues to assist Tip-Adapter and PLOT in achieving improved performance across various shot scenarios in few-shot learning tasks. This underscores the versatility of our proposed framework in addressing more generalized vision reasoning, thereby expanding the applicability of our framework.

Table 7: The few-shot performance of the combination of our method and Tip-adapter-F and PLOT on Caltech101 dataset. The baseline results are from (Chen et al., 2022).

| Methods | 1-shot | 2-shots | 4-shots | 8-shots | 16-shots |
|---|---|---|---|---|---|
| Tip-Adapter-F | 89.33 | 89.74 | 90.56 | 91.44 | 92.86 |
| + OURS | 89.61$_{\uparrow 0.27}$ | 90.98$_{\uparrow 1.24}$ | 91.35$_{\uparrow 0.79}$ | 92.08$_{\uparrow 0.64}$ | 93.27$_{\uparrow 0.41}$ |
| PLOT | 89.83 | 90.67 | 90.80 | 91.54 | 92.24 |
| + OURS | 89.90 $_{\uparrow 0.07}$ | 91.56$_{\uparrow 0.89}$ | 91.20$_{\uparrow 0.40}$ | 92.90$_{\uparrow 1.36}$ | 93.47$_{\uparrow 1.23}$ |

Table 8: Ablation of the necessity of fine-tuning backbone and two-phase training framework on ImageNet-LT on ViT-B/16. *D*- indicates decoupled training between phase A and phase B. *J*- indicates jointly training for both phases.

| Scheme | ImageNet-LT | | | |
|---|---|---|---|---|
| | Many | Medium | Few | Overall |
| zero-shot CLIP | 69.21 | 67.59 | 67.74 | 68.30 |
| *J*-{zero-shot CLIP + OURS} | 74.44 | 76.17 | 70.69 | 74.42 |
| *D*-{zero-shot CLIP + OURS} | 75.26 | 76.55 | 71.15 | 75.32 |
| fine-tuned CLIP | 83.49 | 68.29 | 57.14 | 72.63 |
| *J*-{fined-tuned CLIP + OURS} | 78.26 | 76.24 | 76.09 | 77.00 |
| *D*-{fined-tuned CLIP + OURS} | 81.42 | 77.73 | 71.72 | 78.25 |

## 5.7 Computation Cost Evaluation.

As shown in Table 9, we provide the comparison of computation cost evaluation among BALLAD (Ma et al., 2021), VL-LTR (Tian et al., 2022) and ours. We report the number of training epochs in both phases, training time consumption in Phases B, inference time, the requirement of additional text descriptions, and overall performance. Our method uses acceptable training time to obtain the best performance.

Table 9: Computation cost evaluation on Places-LT based on ViT-B/16. We test all the methods on the same device with a single GPU and batch size is 512. The input size is $3 \times 224 \times 224$. We report the average value of three independent runs and compare the cost from the perspective of Epoch, Training Time (in phase B, minutes), Inference Time (images/s), whether using additional data and Accuracy.

| Method | Epoch | Training Time | Inference Time | Additional Data | Overall Accuracy |
|---|---|---|---|---|---|
| BALLAD | 50+10 | 53.3 | 173.8 | - | 49.5 |
| VL-LTR | 50+50 | 153.4 | 136.7 | ✓ | 50.1 |
| OURS | 50+20 | 109.8 | 158.9 | - | 51.9 |

## 5.8 Others

Please refer to App. D for a deep analysis of our method, including the influence of initialization, batch size, and sampling strategies. We give a visualization of the transport plan to prove its effectiveness in capturing the similarity between visual features and prototypes in App. E and also state the limitations of our method.

## 6 conclusion

In summary, this paper proposes a novel and effective framework for adapting CLIP to long-tailed vision recognition. Based on the observation that fine-tuned textual features are still nearly balanced, we propose an OT-based prototype-guided framework for disentangled vision representation learning. Further, we design a supervised contrastive loss based on the transport plan for further robust visual feature learning. Notably, our method does not require additional data for training and can quickly leverage traditional image-modal methods into the framework for further improvements. Extensive experiments on benchmark datasets prove that our framework can help discriminative visual feature learning and achieve better long-tailed classification performance.

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

## A  Visualization Details about Motivation

For the implementation details of Figure 1, we follow the full fine-tuning of CLIP encoders mentioned in the Section. 3 to make the vision and language encoders of CLIP adapt to the Places-LT training dataset, where the text inputs are constructed by filling `"a photo of a [label]."`. After fine-tuning, we build the vision-language input pairs on the target dataset, where we use 80 prompt sentences for each class to better visualize the textual feature distribution. These sentences are autogenerated on the basis of the prompt templates provided in CLIP. Finally, we use fined-tuned CLIP encoders to extract features and use t-SNE for visualization.

Here, we provide the visualization of 10 classes with the least number of samples based on pre-trained and fine-tuned CLIP, shown in Figure 6 and Figure 7. We can observe that, the imbalanced fine-tuning has little influence on the distribution of visual and textual features, where the visual features are coupled even without fine-tuning. This also demonstrates that pre-trained CLIP without fine-tuning does not have enough capacity to recognize the tail classes of the Places-LT dataset perfectly and we can observe the imbalanced fine-tuning. However, textual features are more discriminative than visual features, which show a potential direction for disentangled learning of balanced vision representation.

We also provide the visualization of 10 classes with the most number of samples based on vanilla pre-trained and fine-tuned CLIP, shown in Figure 8 and Figure 9. We can observe that these classes have a more discriminative and balanced representation distribution than classes with the least samples. This indicates that the encoders have the essential ability to handle head classes, where their representation distributions are relatively disentangled. Therefore, through the bridge of language modality, we can enhance the independence and separability of the vision distribution of the tail class. Ultimately, the discriminability of the distribution of tail classes should be close to that of head classes.

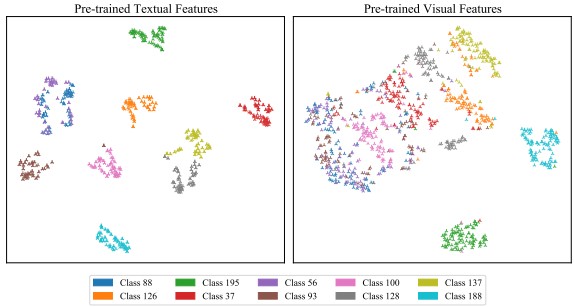

Figure 6: Visualization of pre-trained features from CLIP on 10 classes with **least** number of samples on Places-LT based on ViT-B/16.

Figure 7: Visualization of fine-tuned features from CLIP on 10 classes with **least** number of samples on Places-LT based on ViT-B/16.

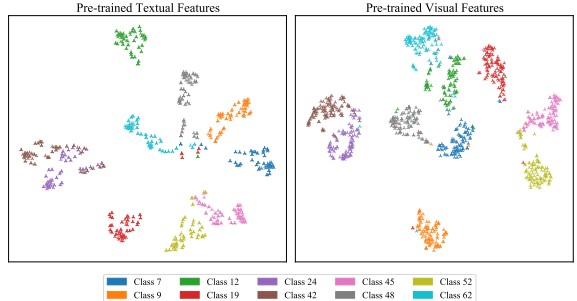

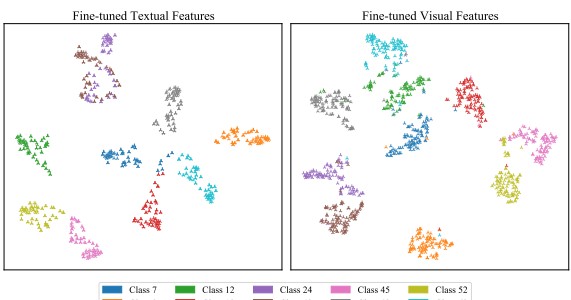

Figure 8: Pre-trained features from CLIP on 10 classes with **most** number of samples.

Figure 9: Fine-tuned features from CLIP on 10 classes with **most** number of samples.

Table 10: Detailed hyper-parameter settings for Phase A and Phase B training on ImageNet-LT, Places-LT and iNaturalist2018. *LR* indicates the learning rate.

| Phase | Dataset | Batch Size | Epoch | LR of encoders | LR of the adapter | LR of the classifier | Momentum | Weight Decay | Scheduler | Sampler |
|---|---|---|---|---|---|---|---|---|---|---|
| A | ImageNet-LT | 512 | 50 | $1e^{-5}$ | - | - | 0.9 | $5e^{-5}$ | cosine | Random |
|   | Places-LT | 512 | 50 | $1e^{-5}$ | - | - | 0.9 | $5e^{-5}$ | cosine | Random |
|   | iNaturalist2018 | 512 | 200 | $1e^{-6}$ | - | - | 0.9 | $5e^{-5}$ | cosine | Random |
| B | ImageNet-LT | 512 | 50 | - | 0.2 | $1e^{-3}$ | 0.9 | $5e^{-5}$ | cosine | Balanced |
|   | Places-LT | 512 | 20 | - | 0.2 | $1e^{-3}$ | 0.9 | $5e^{-5}$ | cosine | Balanced |
|   | iNaturalist2018 | 512 | 50 | - | 0.2 | $1e^{-3}$ | 0.9 | $5e^{-5}$ | cosine | Balanced |

Table 11: Comprehensive comparison results on ImageNet-LT, Places-LT and iNaturalist2018. *Initialization* indicates the initialization weights, where *ImageNet* means the model is pre-trained on the full ImageNet dataset and *CLIP* means the model is initialized by the CLIP weights.

| Methods | Initialization | Backbone | ImageNet-LT | | | | Places-LT | | | | iNat2018 |
|---|---|---|---|---|---|---|---|---|---|---|---|
| | | | Many | Medium | Few | Overall | Many | Medium | Few | Overall | Overall |
| NCM (Kang et al., 2019) | ImageNet | RNXT152 | 60.3 | 49.0 | 33.6 | 51.3 | 40.4 | 37.1 | 27.3 | 36.4 | - |
| | CLIP | RN50 | 58.9 | 46.6 | 31.3 | 49.2 | 37.1 | 30.6 | 19.9 | 30.8 | 65.3 |
| cRT (Kang et al., 2019) | ImageNet | RNXT152 | 64.7 | 49.1 | 29.4 | 52.4 | 42.0 | 37.6 | 24.9 | 36.7 | - |
| | CLIP | RN50 | 63.3 | 47.2 | 27.8 | 50.8 | 38.5 | 29.7 | 17.6 | 30.5 | 69.9 |
| $\tau$-noamrlized (Kang et al., 2019) | ImageNet | RNXT152 | 62.2 | 50.1 | 35.8 | 52.8 | 37.8 | 40.7 | 31.8 | 37.9 | - |
| | CLIP | RN50 | 60.9 | 48.4 | 33.8 | 51.2 | 34.5 | 31.4 | 23.6 | 31.0 | 71.2 |
| LWS (Kang et al., 2019) | ImageNet | RN50 | 57.1 | 45.2 | 29.3 | 47.7 | - | - | - | - | 69.5 |
| | ImageNet | RNXT152 | 63.5 | 50.4 | 34.2 | 53.3 | 40.6 | 39.1 | 28.6 | 37.6 | - |
| | CLIP | RN50 | 62.2 | 48.6 | 31.8 | 51.5 | 36.0 | 32.1 | 20.7 | 31.3 | 71.0 |
| RIDE (4 Experts) (Wang et al., 2021b) | ImageNet | RN50 | 68.2 | 53.8 | 36.0 | 56.8 | - | - | - | - | - |
| | ImageNet | RN50 | 66.2 | 52.3 | 36.5 | 55.4 | - | - | - | - | 72.6 |
| PaCo (Cui et al., 2021) | ImageNet | RNXT101 | 68.2 | 58.7 | 41.0 | 60.0 | 36.1 | 47.9 | 35.3 | 41.2 | - |
| | ImageNet | RN152 | - | - | - | - | - | - | - | - | 75.2 |
| DeiT-B/16 (Touvron et al., 2021) | - | - | - | - | - | - | - | - | - | - | 73.2 |
| LiVT (Xu et al., 2023) | ImageNet | ViT-B/16 | 73.6 | 56.4 | 41.0 | 60.9 | 48.1 | 40.6 | 27.5 | 40.8 | 76.1 |
| LPT (Dong et al., 2022) | ImageNet | ViT-B/16 | - | - | - | - | 49.3 | 52.3 | 46.9 | 50.1 | 76.1 |
| BALLAD (Ma et al., 2021) | CLIP | RN50 | 71.0 | 66.3 | 59.5 | 67.2 | 46.7 | 48.0 | 42.7 | 46.5 | 74.2 |
| | CLIP | ViT-B/16 | 79.1 | 74.5 | 69.8 | 75.7 | 49.3 | 50.2 | 48.4 | 49.5 | - |
| VL-LTR (Tian et al., 2022) | CLIP | RN50 | **77.8** | 67.0 | 50.8 | 70.1 | **51.9** | 47.2 | 38.4 | 48.0 | 74.6 |
| | CLIP | ViT-B/16 | **84.5** | 74.6 | 59.3 | 77.2 | **54.2** | 48.5 | 42.0 | 50.1 | 76.8 |
| OURS | CLIP | RN50 | 72.1 | **70.9** | **63.4** | 70.3 | 47.5 | **49.5** | **50.5** | **49.0** | **75.5** |
| | CLIP | ViT-B/16 | 81.4 | 77.7 | 71.7 | 78.2 | 49.8 | 52.6 | 54.2 | 51.9 | 77.6 |
| OURS + TTA | CLIP | ViT-B/16 | 81.3 | **78.4** | **73.1** | **78.8** | 50.2 | **53.0** | **54.8** | **52.4** | **78.9** |

# B  Implementation details and Detailed information of training hyper-parameters

For data pre-processing, images are resized to $224 \times 224$ and augmented with crop and random horizontal flip following (Ma et al., 2021; Tian et al., 2022). We adopt the class-balanced sampling strategy following (Kang et al., 2019), where the process can be decoupled into two steps-first selecting classes from the list of categories in equal probability and then randomly sampling a data point from the selected class. Unless otherwise specified, the hyper-parameter $\gamma$ for the OT entropy constraint is 0.1 and the maximum iteration number of the Sinkhorn algorithm is 200. For $\tau$, we set it as 0.01 in Phase A and 0.02 in Phase B. For residual factor $\beta$ in the adapter, we set it as 0.2. In Phase B, $\lambda_1 = 1$ and $\lambda_2 = 1$, where ablation studies are provided. All the experiments are conducted on 4 Tesla-A100 GPUs (Phase A) and 4 RTX 4090 GPUs (Phase B). We report the average results by three runs. As shown in Table 10, we give the detailed training hyper-parameters in two phases given three benchmarks for reproducing the results.

# C  Comprehensive comparison with competitive methods

In this section, we give a comprehensive comparison between our method with competitive methods addressing the long-tailed recognition issue. From the Table. 11, we can observe that our method achieves the better performance on three benchmarks under different architectures even without test-time aggregation.

# D Deep Ablation Study

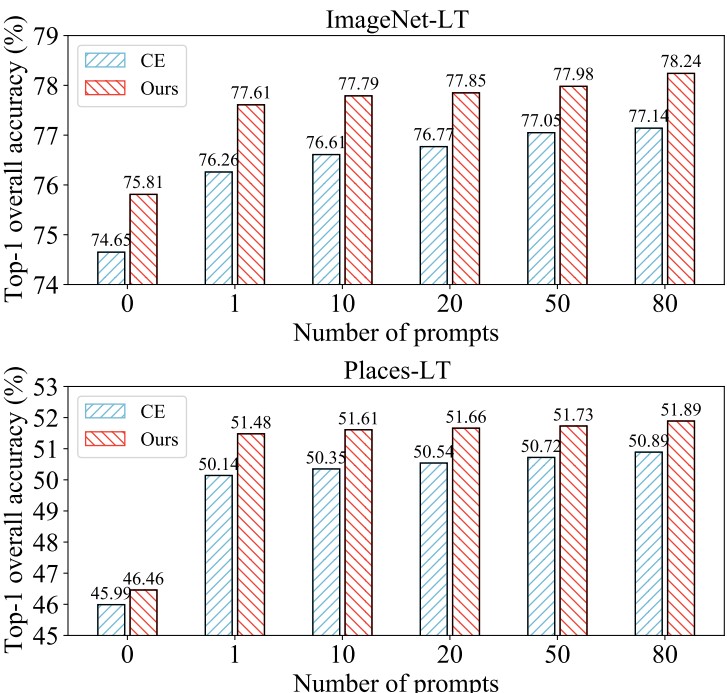

Figure 10: Visualization of the output logtis on ImageNet-LT on ViT-B/16.

## D.1 The influence of the initialization of weights.

Considering the classifier $g$ is initialized by text prompt embeddings, we conduct extensive experiments to investigate how the initialization influences the performance. All the experiments are training for 100 epochs. As shown in Figure 10, the performance of random initialization is significantly lower than that of using prompt embeddings as initialization. In addition, with the help of our proposed loss, the results of random initialization can even be further improved compared to vanilla cross-entropy loss (CE), which also proves the effectiveness of our proposed method in help better representation learning. In addition, beyond using a single prompt embedding as the initialization of the classifier, *i.e.*, `a photo of a [label].`, we also consider how to better utilize the information of multiple prompt embeddings to help classification. From the representation perspective, we hope that the learned visual features can be close to multiple prompt embeddings at the same time. In practice, we simply average the multiple prompt embeddings of each class and then use them as the initialization of the classifier $g$. As shown in the second group, overall performance has been improved by 0.36% in ImageNet-LT and 0.39% in Places-LT with the increasing of number of prompts from 1 to 80. This shows that our method can also leverage multiple text prompts for better classification performance. More value details can be found in Table. 12.

## D.2 The influence of batch size

Considering the computation of the proposed $\mathcal{L}_{\text{OT}}$ in Eq.6 is influenced by the input batch size $B$, we conduct an ablation study on it to examine how the batch size affects our method. We use a batch size of 512 as a benchmark, which corresponds to an adapter learning rate of 0.2 and a classifier learning rate of 0.001. When the batch size increases or decreases, the learning rate will also proportionally increase or decrease. As shown in Table 13, setting batch size as 512 gives the best performance.

Table 12: Ablation on number of prompts for initialization of classifier $g$ on ImageNet-LT and Places-LT based on ViT-B/16. $A$ indicates the number of prompts and $B$ indicates CE (C) or OURS (O) for *"A - B"*, where $A = 0$ means we randomly initialize the classifier $g$, $A = 1$ means we use the embedding of *"a photo of {classname}"* for the initialization and the others means we use the average of the corresponding number of prompt embeddings.

| Number of Prompts | ImageNet-LT | | | | Places-LT | | | |
|---|---|---|---|---|---|---|---|---|
| | Many | Medium | Few | Overall | Many | Medium | Few | Overall |
| 0 - C | 78.07 | 74.75 | 64.63 | 74.65 | 46.05 | 45.99 | 46.30 | 45.88 |
| 0 - O | 78.88 | 75.91 | 65.60 | 75.81 ↑1.16 | 47.11 | 46.66 | 44.80 | 46.46 ↑0.58 |
| 1 - C | 75.82 | 75.90 | 78.76 | 76.26 | 48.59 | 50.67 | 53.18 | 50.14 |
| 1 - O | 80.37 | 77.44 | 74.65 | 77.61 ↑1.35 | 47.96 | 52.37 | 55.92 | 51.48 ↑1.34 |
| 10 - C | 76.47 | 76.44 | 77.59 | 76.61 | 48.61 | 50.81 | 53.67 | 50.35 |
| 10 - O | 80.45 | 77.56 | 74.45 | 77.79 ↑1.18 | 49.26 | 52.02 | 55.00 | 51.61 ↑1.26 |
| 20 - C | 76.63 | 76.70 | 77.43 | 76.77 | 48.57 | 50.64 | 53.96 | 50.54 |
| 20 - O | 80.37 | 77.56 | 74.54 | 77.85 ↑1.08 | 49.19 | 52.16 | 55.09 | 51.66 ↑1.08 |
| 40 - C | 77.26 | 76.96 | 76.76 | 77.05 | 48.68 | 50.79 | 54.34 | 50.72 |
| 40 - O | 80.47 | 77.55 | 74.43 | 77.98 ↑0.93 | 49.84 | 52.39 | 53.72 | 51.73 ↑1.01 |
| 80 - C | 77.43 | 77.19 | 76.34 | 77.14 | 48.73 | 51.21 | 54.65 | 50.89 |
| 80 - O | 82.09 | 77.65 | 72.34 | 78.25 ↑1.10 | 49.82 | 52.56 | 54.20 | 51.89 ↑1.00 |

Table 13: Ablation on training batch size of Phase B on ImageNet-LT and ViT/B-16.

| Batch Size | ImageNet-LT | | | |
|---|---|---|---|---|
| | Many | Medium | Few | Overall |
| 128 | 81.16 | 77.27 | 72.54 | 78.20 |
| 256 | 81.27 | 77.08 | 72.88 | 78.23 |
| 512 | 81.09 | 77.65 | 72.34 | 78.24 |
| 1024 | 80.68 | 77.45 | 72.84 | 78.23 |

## D.3 Performance of different sampling strategies

In addition, we delve into various sampling strategies such as class-balanced sampling, square root sampling, and mix-balanced sampling for Phase B. Class-balanced sampling selects categories from the original dataset with equal probability, unlike the natural instance-balanced sampling that chooses instances irrespective of their classes. This process can be broken down into two steps - first, classes are selected equally from the list of categories, and then a data point is randomly sampled from the chosen class. Square-root sampling (Mahajan et al., 2018) initially calculates the square root of the number of head classes, then re-normalizes and carries out sampling based on the resulting distribution. Mix-balanced sampling merges instance-balanced sampling and class-balanced sampling, thereby leveraging both strategies to prevent overfitting in early epochs and underfitting in late epochs. Inspired by (Kang et al., 2019), we employ a soft version of mix-balanced sampling to dynamically interpolate between instance-balanced sampling and class-balanced sampling as learning advances. As demonstrated in Table 14, class-balanced sampling can significantly aid medium-shot and few-shot categories. Therefore, we utilize class-balanced sampling as the balancing method of BALLAD.

## D.4 The influence of different metrics in supervised contrastive loss

To demonstrate the efficiency of the proposed supervised contrastive loss based on the transport plan, we compare different metrics serving the distance measurement in Eq. 7 on ImageNet-LT based on ViT-B/16. As shown in Table. 15, the first group indicates that we use the logits from the classifier $g$ or cost function

Table 14: Comparison of different balanced sampling strategies on ImageNet-LT on ViT/B-16.

| Balance method | ImageNet-LT | | | |
| --- | --- | --- | --- | --- |
| | Many | Medium | Few | Overall |
| Class-Balanced | 81.42 | 77.73 | 71.72 | 78.25 |
| Square-root | 80.83 | 76.87 | 70.32 | 78.00 |
| Mix-balanced | 81.63 | 77.28 | 70.74 | 78.13 |

**C** in Eq. 6 to compute $\mathcal{L}_{\text{SCT}}$ instead of $\mathbf{T}^*$. We can observe that when employing logits from $g$ to compute the loss, the model will collapse and performance is going to degenerate to 0. Maybe it is incurred that the logits will used to compute both $\mathcal{L}_{\text{CS}}$ and $\mathcal{L}_{\text{SCT}}$, and the inconsistency push model collapse. When we use **C** with $l_1$ distance to compute $\mathcal{L}_{\text{SCT}}$, we can obtain an sub-optimal performance. The second group shows the influence of using $l_1$ distance, $l_2$ distance and Dot distance to implement the $\mathcal{L}_{\text{SCT}}$ based on the optimal transport plan $\mathbf{T}^*$. We can observe that the $l_1$ gives the best performance.

Table 15: Comparison of different metrics of Eq. 7 on ImageNet-LT on ViT-B/16.

| Metrics | ImageNet-LT | | | |
| --- | --- | --- | --- | --- |
| | Many | Medium | Few | Overall |
| Logits + $l_1$ | 0 | 0.01 | 0 | 0 |
| **C** + $l_1$ | 81.14 | 77.55 | 71.62 | 78.13 |
| $l_1$ distance | 81.42 | 77.73 | 71.72 | 78.25 |
| $l_2$ distance | 81.59 | 77.33 | 71.07 | 78.12 |
| Dot distance | 81.61 | 77.42 | 70.99 | 78.16 |

### D.5  When to employ balanced sampling strategy?

As shown in Table 16, we provide the ablation study on where to employ the balanced sampling strategy. We can observe that balanced sampling only in Phase B gives the best performance. In addition, in Figure 11, we can see that even when balanced sampling strategy is used in Phase A, it cannot help tail classes obtain a more discriminative visual feature distribution. The visual feature distribution obtained by balance sampling does not have a significant difference from that obtained by random sampling on tail samples. However, for head samples shown in Figure 12, we can see that the visual features of the head class after balance sampling fine-tuning are clustered less tightly than those obtained by random sampling. This may also explain why the performance of using balance sampling in both phases is not as good as using it only in Phase B, especially for the results that the performance of Many-shot for using balanced sampling in both phases is lower than only in Phase B (48.9% vs 49.8%). Therefore, using balance sampling in Phase B gives the best performance.

Table 16: Which training phase to employ balanced sampling strategy ablations. On the Places-LT dataset, balanced sampling only in Phase B makes our method perform best.

| Dataset | Balance Sampling | | Many | Medium | Few | Overall |
| --- | --- | --- | --- | --- | --- | --- |
| | Phase A | Phase B | | | | |
| Places-LT | - | - | **55.1** | 44.0 | 38.2 | 44.9 |
| | ✓ | - | 54.4 | 43.1 | 39.7 | 45.2 |
| | ✓ | ✓ | 48.9 | 52.1 | 52.3 | 51.0 |
| | - | ✓ | 49.8 | **52.6** | **54.2** | **51.9** |

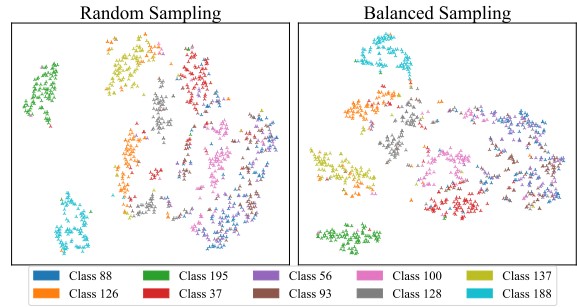
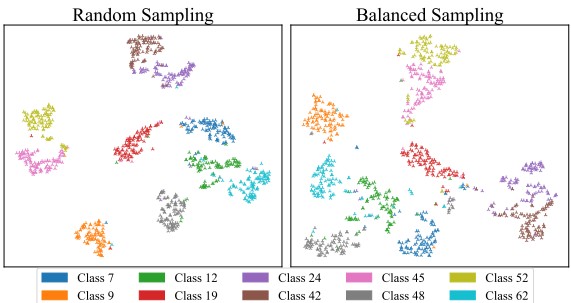

Figure 11: Fine-tuned visual features from CLIP on 10 classes with **least** number of samples on Places-LT on ViT-B/16.

Figure 12: Fine-tuned visual features from CLIP on 10 classes with **most** number of samples on Places-LT on ViT-B/16.

### D.6 Implementation details about how to combine our framework with vision-modality-based methods

All the experiments share the same Phase A for fine-tuning encoders. In combination with CB-CE loss and TTA, we train the model from the stretch in Phase B. In combination with LWS and MisLAS, we use our best checkpoint in Phase B as the initialization and then employ these methods for further tuning.

**Combination with CB-CE loss.** We simply replace our $\mathcal{L}_{\mathrm{CE}}$ with CB-CE loss in our $\mathcal{L}_{\mathrm{B}}$ to train the models from scratch, where the hyper-parameter $\beta$ in CB-CE is 0.99.

**Combination with LWS.** We first initialize the model with our best checkpoint, where the encoders and the linear adapter are all frozen. Then we employ LWS to further tune the classifier with learnable parameters to adjust the loss weights. We use $\mathcal{L}_{\mathrm{CE}} + \mathcal{L}_{\mathrm{OT}} + \mathcal{L}_{\mathrm{SCT}}$ to optimize the classifier and learnable parameters.

**Combination with MisLAS.** We use LabelSmoothLoss proposed in original MisLAS to replace $\mathcal{L}_{\mathrm{CE}}$ for tuning the classifier. Other settings are the same as the combination with LWS.

**Combination with TTA.** We only aggregate the logits of the classifier from five perturbed copies of the input image to evaluate the model and keep the same training strategy as our proposed method for model learning.

## E Visualization of Transport Plan

In this section, we give the visualization of the transport plan on the test dataset of ImageNet-LT based on ViT-B/16. The transport plan captures the similarity between $p$ and $q$. In our work, we utilize the transport plan to push refined visual features close to the corresponding prototypes. Therefore, the visualization of the transport plan can visually prove the rationality and effectiveness of our method, by using optimal transport to measure the distance between refined visual features and prototypes.

As shown in Figure 13, the horizontal axis is the weights $\mathbf{W}$ of the classifier $g$, sorted by index. The vertical axis is the test set, sorted by label. The perfect transport plan should be diagonal because we initialize $\mathbf{W}$ using textual features corresponding to images with the same label. Namely, $\mathbf{W}_{g,1}$ is initialized by the textual features of $1-$st class. Therefore, $\mathbf{W}_{g,1}$ should be more similar to the visual features of $1-$st class. As can be seen, our method keeps the diagonal, showing that the transport plan can effectively capture the similarity between refined visual features and prototypes. As a result, our proposed method based on optimal transport is reasonable.

## F Limitations

Similar to BALLAD and VL-LTR, Our proposed method requires fine-tuning the visual encoder in downstream datasets, which require some computational resources and may limit our method. In the future, we

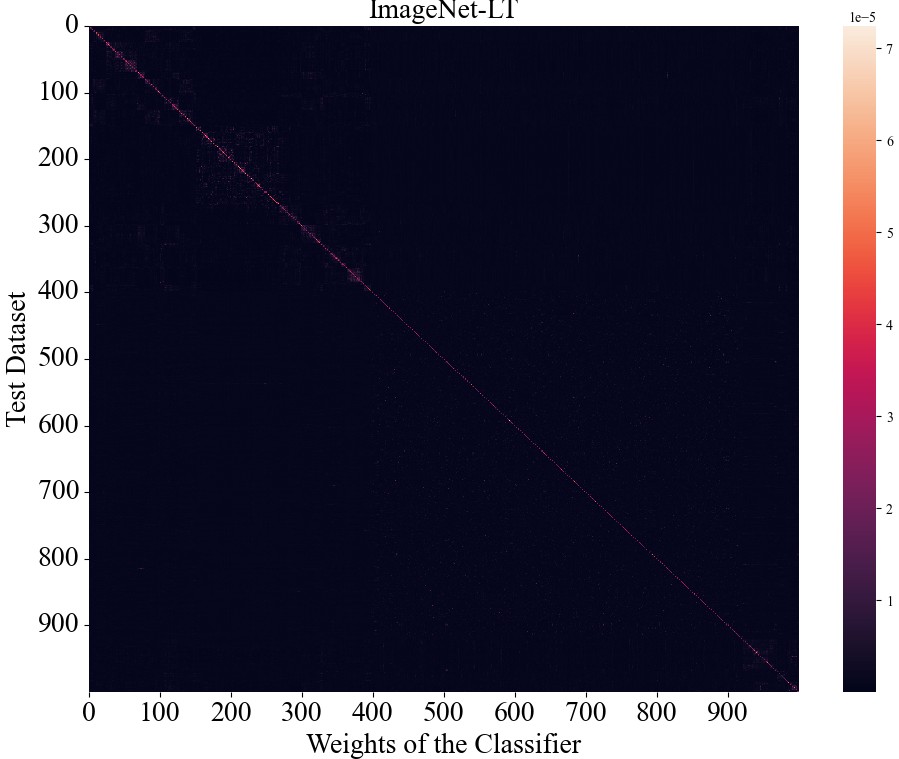

Figure 13: Visualization of the transport plan on ImageNet-LT on ViT-B/16.

will focus on how to leverage some parameter-efficient tuning methods into our framework to reduce the requirement of cost.

