# OpenReview forum: "Textual Prototypes Guided Balanced Visual Feature Learning For Long-Tailed Vision Recognition"
_TMLR — Rejected by TMLR_

### Review · Reviewer_cyMU · 2025-04-20

**Summary Of Contributions:**

The paper proposes a robust CLIP-based training method to address the long-tailed vision recognition challenge. It leverages balanced textual features as prototypes to guide the learning of robust and disentangled representations from biased visual features via using optimal transport (OT) to align visual features with textual prototypes. Experiments demonstrate significant improvements over previous methods.

**Audience:**

Yes

**Broader Impact Concerns:**

No concerns.

**Claims And Evidence:**

Yes

**Requested Changes:**

I am not familiar with the  OT‑based CLIP strategy, but it would greatly strengthen the paper to include a thorough comparison with other methods that apply optimal transport within CLIP fine‑tuning (A-C). Specifically, please discuss how your approach differs in formulation, training cost, and empirical performance from prior OT‑augmented CLIP training techniques.

[A]  OT-CLIP: Understanding and Generalizing CLIP via Optimal Transport, 2024 ICML

[B] PLOT: Prompt Learning with Optimal Transport for Vision-Language Models, 2023 ICLR

[C] Global and Local Prompts Cooperation via Optimal Transport for Federated Learning, CVPR  2024

**Strengths And Weaknesses:**

**Summary of strengths**

- The paper is well organized, clear, and easy to follow.
- It tackles a method for the challenging of long‑tailed visual recognition, which remains highly relevant in real‑world applications.
- Extensive experiments and ablation studies consistently demonstrate performance gains over baselines.

**Summary of weaknesses**

- The methodological novelty is limited:  using textual features (optimal transport) to guide visual representations is broadly applicable and does not exploit. The key components (e.g. the adapter‑based tuning, prototype initialization, and OT alignment) largely extend existing ideas, and the overall improvements are modest.
- Lacks theoretical justification or formal guarantees for the proposed framework can address the long-tailed problem.

---

### Review · Reviewer_2v69 · 2025-05-18

**Summary Of Contributions:**

To address the visual long-tail recognition problem, this paper first proposes using a pre-trained CLIP model and conducting CLIP fine-tuning on image-text pairs with a lightweight vision adapter model. Then, the authors propose fine-tuning the visual representation using an additional linear classifier that maps the visual encoding of CLIP to the k classes, initializing the classifier’s weight matrix with the k text prototype vectors from the pre-trained vision-language CLIP model. The authors further propose using optimal transport to perform unsupervised prototype-guided learning, aligning the distribution of visual features with the k classes. Additionally, they introduce a supervised contrastive constraint on the transport plan to encourage visual features from the same class to be close together, and those from different classes to be farther apart. They conduct experiments on ImageNet and Places-LT datasets, along with corresponding ablation studies. Their model outperforms vision recognition models up to 2022, particularly on the tail classes in the long-tail recognition problem. Their qualitative analysis and visualizations also show that the proposed methods can disentangle the classes in the representation space.

**Audience:**

Yes

**Broader Impact Concerns:**

No specific concerns.

**Claims And Evidence:**

Yes

**Requested Changes:**

Please refer to the weakness section.

In addition, please refer to the minor comments below:

Minor Comments:
1.	are a few typos in the paper. For example, in Section 6, the word Conclusion should have the “C” capitalized.
2.	The cosine similarity between the text and image modalities is a key aspect of the proposed method. While this is emphasized in the paper, it would be helpful to provide more visualizations or application studies related to this, in addition to what is shown in Figure 5.
3.	The mathematical notations are generally consistent. However, in Section 4.2, it would be helpful if the authors could briefly explain the intuition and mathematical meaning behind initializing the weight matrix $W_g$ using the text prototypes. Specifically, why would this kind of initialization lead to a better and more disentangled classifier? This is not obvious from the current explanation to me.

**Strengths And Weaknesses:**

Strengths:
	1.	The paper is generally well written and easy to follow. The mathematical notations are consistent, well defined, and clearly presented.
	2.	The proposed method integrates optimal transport-based algorithms into the long-tail image classification problem, following the pre-training and fine-tuning of the CLIP model on imbalanced class distributions.
	3.	The visualization and qualitative analysis demonstrate that the proposed method can effectively separate and disentangle visual features. Additionally, the inclusion of optimal transport and prototype matching plays an important role in addressing the long-tail recognition challenge.
	4.	The quantitative results show that the model outperforms previous methods, particularly on the under-represented classes in the ImageNet-LT and Places-LT long-tail datasets.


Weaknesses:
	1.	All the compared baseline methods, such as those listed in Table 1, Table 2, and Table 5, are relatively outdated. The most recent models included for comparison are from 2022, with no inclusion of more recent long-tail recognition models from 2023 or 2024 throughout the paper’s analysis.
	2.	The proposed use of optimal transport and the additional contrastive supervised constraint on the transport plan appears quite similar to the original CLIP loss, as both aim to perform contrastive learning based on cosine similarity between image and text modalities. Moreover, the necessity of introducing these components is not clearly supported by the relatively marginal improvement of ablations in the experimental results.
	3.	In Sections 4.2 and 4.3, the text prototype-guided feature learning via optimal transport and the contrastive supervised constraint is described, and the ablation study shows that these components provide only marginal performance gains. It remains unclear whether similar improvements could be achieved using the standard CLIP loss with balanced fine-tuning as in Phase B.
	4.	Furthermore, although the proposed method brings substantial gains in underrepresented classes (Table 1), it also results in noticeable performance regression—up to 6%—on the well-represented many-shot classes in the ImageNet-LT dataset.

---

### Review · Reviewer_obnY · 2025-06-02

**Summary Of Contributions:**

The paper presents an approach for large-scale image classification, focusing particularly on improving classification accuracy on rarer image classes. To this end, the authors first experimentally observe that the textual features of CLIP are more well-separated than the corresponding visual features. They leverage this observation to design a method comprising (first) fine-tuning CLIP features on a given dataset, and (then) training linear adapters to further refine the visual features through a classifier network whose weights are initialized to be the fine-tuned CLIP text features. For datasets that do not contain text captions of images, the authors construct text captions of the style $\texttt{a photo of a [label]}$, with $\texttt{label}$ being the name of the image class. The authors experimentally show that the resultant refined visual features are better separated for robust classification than the fine-tuned visual features. They also report the classification accuracies and computational costs of their proposed method on three open-source datasets for many-shot ($\geq 100$ samples), medium-shot ($20 \sim 100$ samples), and few-shot ($\leq 20$ samples) classification tasks, and demonstrate the contributions of their proposed components through ablation studies.

**Audience:**

Yes

**Claims And Evidence:**

Yes

**Requested Changes:**

1. Please address the concerns mentioned under "Weaknesses".

2. The paper's title specifically mentions "balanced" visual feature learning, but the balancing is only performed through an existing balanced sampling method and is seemingly not a contribution of the work. Is including "balanced" in the title -- which indicates to the reader that balancing is a contribution -- still appropriate?

3. Including any relevant information on specific improvements on the long tail (rarer classes) in the abstract and the introduction can help tie together the key claims and experimental takeaways of the paper.

4. In the Introduction, the authors mention "five approaches" for improving long-tailed recognition. But the last one is called "others", which is not an approach in the same sense as the other four. The authors may want to revisit this categorization to make it more rigorous.

5. For completeness, it may be worth mentioning that $\sum_{k=1}^K N_k = N$ in page 3, Section 3, para 1.

6. Page 4, Section 3, second-last para: "a set of text prompts $\mathbf{t}$ with the length $K$" $\rightarrow$ may be clearer to say "of cardinality $K$" since "length" can have multiple meanings for text prompts.

7. The described imbalance factor is oversensitive to the size of the smallest class and also does not give a clear indication of the long-tail nature of the dataset. For example, a dataset with $K-1$ classes of the same size and only the $K$-th class being the smallest one would have a high IF without having a long-tailed distribution. A better indicator of the true imbalance would be the ratio of the number of samples in the largest class and the number of samples in the median class.

8. Please clarify whether the t-SNEs in Figures 1, 4, and 5 are for the rarest classes.

9. Also, in all the Figures, it would significantly improve readability and understanding to mention class names instead of class numbers.

10. Typos and formatting.

    10.1. It would help readability to enumerate the main contributions in the Introduction.

    10.2. In Figure 2, I would recommend moving the legend at the top-left to the bottom of the figure and laying it out horizontally to make better use of the horizontal space. The current positioning is somewhat unintuitive, given that the legends are for all parts of the figure. It also interferes with the readability of Phase A.

    10.3. I would recommend interchanging the numbering of Tables 7 and 8, since Table 8 is referenced before Table 7.

    10.4. Figure 13 might be easier to read if the color palette goes from light to dark instead of from dark to light.

    10.5. Missed capitalizations: titles of Sections 3 and 6.

    10.6. In Section 3, paragraph "Revisit to CLIP": "align the the visual..." $\rightarrow$ extra "the".

    10.7. In Section 4.2, paragraph "Linear adapter for refining visual features": "$f_{vis}^{a}$ short" $\rightarrow$ "$f_{vis}^{a}$ for short".

    10.8. In Section 4.2, paragraph "Linear adapter for refining visual features": "$\mathbf{v}$ keep unchanged" $\rightarrow$ "we keep $\mathbf{v}$ unchanged".

    10.9. In Algorithm 1, "Section 22" on Line 16 appears to be a typo.

    10.10. In Algorithm 1, Line 18: "Eq. 5" $\rightarrow$ "Eq. 6".

**Strengths And Weaknesses:**

### Strengths
1. The proposed approach of refining visual CLIP features using a classifier with weights initialized with the fine-tuned CLIP text features is sound and clearly explained.
2. The marginal contributions of the proposed linear adaptive layer, and the proposed uses of optimal transport loss and the contrastive supervised constraint on the transport plan are demonstrated through ablation studies.
3. The classification accuracy numbers are complemented with corresponding computational costs of the different methods, thus providing a more holistic idea of the performance of the different methods.
4. Additional experiments, such as training a linear adaptive layer for text features, improve the completeness of the work and further support the proposed claims.

### Weaknesses
1. While the authors show t-SNE plots for the refined visual features for the rarest 10 classes in the datasets, it would be insightful to note the accuracy numbers and confusion matrix for the rarest classes as well. Since the proposed approach claims to improve classification performance on the rarer classes, these numbers should best highlight the key claim of improving classification performance on the rarer classes.

2. One of the author's key claims -- that the CLIP text features are more well-separated than the visual features -- can be better explored. What are the assumptions on the given text for which this observation is true, and is it feasible to provide confidence bounds to support such assumptions? For example, what would the distribution of the fine-tuned text features look like for which they are more well-separated than the corresponding fine-tuned visual features with 95% confidence? This will also better motivate the authors' choice of the constructed text caption $\texttt{a photo of a [label]}$, with $\texttt{label}$, and can provide general guidance on writing text captions best suited for long-tailed image classification.

3. Some aspects of the quantitative performance of the different methods are not fully explained. Why does the accuracy of the proposed method drop below that of VL-LTR for many-shot classification? Is it just because of the additional data VL-LTR uses? Also, without looking at the variances corresponding to the given mean accuracy numbers, and also noting the number of trials to get the means and the variances, it is hard to evaluate the statistical significance of the accuracy improvements (especially as most of the improvements are under 2%).

4. The quantitative results can be better supported with qualitative comparisons and perceptual studies. For example, providing side-by-side examples of images correctly classified by the proposed method and incorrectly classified by the other methods, and providing supporting visuals, such as saliency maps, can clearly highlight any practical performance benefits of the proposed approach. Further, conducting perceptual studies (asking humans to label the images) on the marginal examples where only the proposed method performs correctly can establish a sense of "hardness" of these marginal examples. Such qualitative evaluations will provide a well-rounded idea of the benefits of the proposed method beyond the oversimplified mean accuracy numbers.

---

### Decision · Action_Editor_RVP4 · 2025-07-23

**Recommendation:** Reject

**Additional Comments:**

Unfortunately, the authors did not provide any response on these points.

Therefore, I would recommend the paper to be rejected.

**Audience:**

No

**Audience Explanation:**

In addition, the lack of experimental comparison with more recent methods (2022~) further reduces the interest for TMLR's audiences.

**Claims And Evidence:**

No

**Claims Explanation:**

This paper proposes a robust visual feature learning guided by textual features from CLIP for long-tailed image classification. In particular, in order to obtain more separated and discriminative visual features for all classes including rare classes, after fine-tuning visual and textual features from CLIP on a target imbalanced task, it introduces a linear adapter on visual features and a linear classifier initialized by textual features from class-specific text prompts and learns by the cross-entropy (CE) loss, optimal transport (OT)-based prototype-guided loss, and supervised contrastive (SCT) loss.

Overall, the paper is well organized and easy to understand. The proposed method is somewhat technically sound. However, the claim that the proposed optimal transport (OT)-based prototype-guided loss and the supervised contrastive (SCT) loss lead to disentangled and robust vision representation learning for improved few-shot classification is not sufficiently supported by experimental results. For example, in Table 3, the use of the OT-loss and SCT loss leads to only marginal improvements in accuracy. Moreover, their impact on few-shot classes is not clearly demonstrated. As shown in Table 1, the proposed method even degrades performances on many-shot classes.